# Impact of early corticosteroids on 60-day mortality in critically ill patients with COVID-19: A multicenter cohort study of the OUTCOMEREA network

Claire Dupuis[1,2]*, Etienne de Montmollin[2,3], Niccolò Buetti[3], Dany Goldgran-Toledano[4], Jean Reignier[5], Carole Schwebel[6], Julien Domitile[1], Mathilde Neuville[7], Moreno Ursino[8,9], Shidasp Siami[10], Stéphane Ruckly[11], Corinne Alberti[12], Bruno Mourvillier[13], Sebastien Bailly[14], Virginie Laurent[15], Marc Gainnier[16], Bertrand Souweine[1], Jean-François Timsit[2,3], on behalf of the OutcomeRea™ research network¶

1 Medical Intensive Care Unit, Gabriel Montpied University Hospital, Clermont-Ferrand, France, 2 Université de Paris, UMR 1137, IAME, Paris, France, 3 APHP, Medical and Infectious Diseases Intensive Care Unit, Bichat-Claude Bernard Hospital, Paris, France, 4 Polyvalent ICU, Groupe Hospitalier Intercommunal Le Raincy Montfermeil, Montfermeil, France, 5 Medical Intensive Care Unit, Nantes University Hospital, Nantes, France, 6 Medical Intensive Care Unit, Grenoble University Hospital, La Tronche, France, 7 Polyvalent ICU, Hôpital Foch, Suresnes, France, 8 F-CRIN PARTNERS platform, AP-HP, Université de Paris, Paris, France, 9 Centre de Recherche des Cordeliers, INSERM, Sorbonne Université, USPC, Université de Paris, Paris, France, 10 Polyvalent ICU, Centre Hospitalier Sud Essonne Dourdan-Etampes, Dourdan, France, 11 Department of Biostatistics, ICUREsearch, Paris, France, 12 APHP, Epidemiology, Hôpital Robert Debré, Paris, France, 13 Medical Intensive Care Unit, Robert Debré University Hospital, Reims, France, 14 Université Grenoble Alpes, Inserm, CHU Grenoble Alpes, HP2, Grenoble, France, 15 Medical-Surgical Intensive Care Unit, André Mignot Hospital, Le Chesnay, France, 16 APHM, Intensive Care Unit, La Timone University Hospital, Marseilles, France

¶ Membership of The OutcomereaTM study group—outcomerea.fr is listed in the Acknowledgments.
* cdupuis1@chu-clermontferrand.fr

**Data Availability Statement:** All relevant data are within the manuscript and its Supporting Information files.

## Abstract

### Objectives

In severe COVID-19 pneumonia, the appropriate timing and dosing of corticosteroids (CS) is not known. Patient subgroups for which CS could be more beneficial also need appraisal. The aim of this study was to assess the effect of early CS in COVID-19 pneumonia patients admitted to the ICU on the occurrence of 60-day mortality, ICU-acquired-bloodstream infections (ICU-BSI), and hospital-acquired pneumonia and ventilator-associated pneumonia(HAP-VAP).

### Methods

We included patients with COVID-19 pneumonia admitted to 11 ICUs belonging to the French OutcomeReaTM network from January to May 2020. We used survival models with ponderation with inverse probability of treatment weighting (IPTW).

### Results

The study population comprised 303 patients having a median age of 61.6 (53–70) years of whom 78.8% were male and 58.6% had at least one comorbidity. The median SAPS II was

**Funding:** ICUREsearch provided support in the form of salary for SR. The specific roles of these authors are articulated in the 'author contributions' section. The funders had no role in study design, data collection and analysis, decision to publish, or preparation of the manuscript.

**Competing interests:** The authors have read the journal's policy and have the following competing interests: SR is a paid employee of ICUREsearch. There are no patents, products in development or marketed products associated with this research to declare. This does not alter our adherence to PLOS ONE policies on sharing data and materials.

33 (25–44). Invasive mechanical ventilation was required in 34.8% of the patients. Sixty-six (21.8%) patients were in the Early-C subgroup. Overall, 60-day mortality was 29.4%. The risks of 60-day mortality ($_{IPTW}$HR = 0.86;95% CI 0.54 to 1.35, p = 0.51), ICU-BSI and HAP-VAP were similar in the two groups. Importantly, early CS treatment was associated with a lower mortality rate in patients aged 60 years or more ($_{IPTW}$HR, 0.53;95% CI, 0.3–0.93; p = 0.03). In contrast, CS was associated with an increased risk of death in patients younger than 60 years without inflammation on admission ($_{IPTW}$HR = 5.01;95% CI, 1.05, 23.88; p = 0.04).

## Conclusion

For patients with COVID-19 pneumonia, early CS treatment was not associated with patient survival. Interestingly, inflammation and age can significantly influence the effect of CS.

## Introduction

Around five percent of COVID-19 patients developed a severe form of the disease and required intensive care unit (ICU) admission [1, 2]. Inflammation and cytokine storm were observed in most of these ICU patients [3, 4] prompting the investigation of corticosteroids (CS) as a therapeutic option [5].

CS were widely used during the outbreaks of severe acute respiratory syndrome (SARS)-CoV1 and Middle East respiratory syndrome (MERS)-CoV. However, studies yielded conflicting results, with some observational data suggesting increased mortality and secondary infection rates and impaired clearance of SARS-CoV and MERS-CoV [5, 6].

In other clinical settings, studies reported beneficial effects of CS in septic shock [7, 8], and ARDS [9]. However, the results are not generalizable to COVID-19 patients owing to the low frequency of septic shock most of the time and because the ARDS phenotype is quite different [10].

Several randomized controlled trials (RCT) dealing with the impact of CS on COVID-19 patients have been published recently [11–15]. Most of them were collected in a meta-analysis by the REACT working group [16], which found a reduction in 28-day mortality in patients on CS. However, this result should be interpreted with caution for several reasons [17]. First, the study was based mainly on the UK Recovery trial, a large open-label RCT which found that treatment with dexamethasone (6 mg/d for 10 days) reduced mortality. Unfortunately, this result was only preliminary, the trial had a limited follow-up period of 28 days, the pragmatic design did not allow a strict balance between groups, and the adverse effects of CS were not monitored [11]. Second, all other studies included in the meta-analysis were stopped prematurely because of the Recovery trial results and were consequently underpowered with no significant endpoint results assessed before day 28 [12–14]. The final report of the Recovery trial finally confirmed the previous results [18]. The subsequent MEtCOVID trial did not show a difference in mortality at day 28 between treatment groups [15] but suggested a benefit of CS in patients aged over 60. Finally, several observational studies yielded conflicting results [19–23].

In light of these considerations, no definitive conclusion should be drawn [24] and some issues are still pending such as the dosage and timing of CS. Furthermore, as one size dose rarely fits all, some subgroups of patients might benefit from CS more than others, including older patients, more severe ones or those with inflammation.

Against this background, the analysis of observational longitudinal studies could be a suitable alternative to randomized control trials [25, 26] and provide a truer picture of what impact certain measures have.

The aim of this study was to assess in critically ill COVID-19 patients the effect of early CS administration on 60-day mortality, ICU-bloodstream infections (ICU-BSI) and hospital-acquired pneumonia and ventilator-associated pneumonia (HAP-VAP) in various subgroups using a large multicentric observational cohort and applying an inverse probability of treatment weight (IPTW). Cox survival model.

## Materials and methods

### Data source

This study was performed with data from the French prospective multicenter (n = 11 ICUs) OutcomeRea^TM database. The methods for data collection and quality of the database have been described in detail elsewhere [27]. In accordance with French law, the OutcomeRea^TM database has been approved by the French Advisory Committee for Data Processing in Health Research (CCTIRS) and the French Informatics and Liberty Commission (CNIL, registration no. 8999262). The database protocol was approved by the Institutional Review Board of the Clermont-Ferrand University Hospital (Clermont-Ferrand, France), who waived the need for informed consent (IRB no. 5891).

### Study population

Patients over 18 years were eligible for inclusion in the analysis if they were admitted to one of the ICUs belonging to the OutcomeRea^TM network and if they developed a severe COVID-19 disease confirmed by a positive SARS-CoV-2 test using reverse-transcriptase polymerase chain reaction (PCR).

Patients were excluded if they were referred from another ICU, if a decision was made to discontinue life-sustaining treatments during the first two days after ICU admission, if their ICU length of stay was ≤ 2 days and if they had previously received CS before ICU admission.

### Data collection

All data were prospectively collected and comprised details on ICU admission (demographics, chronic disease/comorbidities as defined by the Knaus Scale [28], baseline severity indexes: SAPS II [29] and SOFA [30] scores, treatments on admission including lopinavir-ritonavir, hydroxychloroquine, tocilizumab, Anakinra and CS); several variables recorded throughout the ICU stay (clinical and biological parameters, requirement for non-invasive ventilatory support and invasive mechanical ventilation (IMV) and other organ support [vasopressors, renal replacement therapy]); and outcomes (occurrence of HAP-VAP and ICU-BSI and ICU and hospital length of stay [LOS], vital status at ICU and hospital discharge and at day 60 after ICU admission).

### Definitions, group assignment

The Early-CS group comprised all patients who received corticosteroids for the first time during the first two days after ICU admission. The Non-early CS group included patients who did not receive steroids during the first two days after ICU admission. High doses of corticosteroids concerned patients receiving more than 10 mg of dexamethasone, and more than 200 mg of hydrocortisone. Inflammation was defined by at least two of the following criteria : Ferritin >1000 μg/l or D-Dimers >1000 μg/l or C-reactive protein (CRP) >100 mg/dL [31].

According to the MetCOVID results [15], we also planned to evaluate the subgroups of patients aged over and under 60.

The presence or absence of HAP-VAP and ICU-BSI was documented according to the standard definitions developed by the Centers for Disease Control and Prevention [32]. Quantitative cultures of specimens were required to diagnose HAP-VAP or ICU-BSI.

Length of ICU and hospital stays was calculated from ICU admission.

The positive results of blood culture, pathogen identification and their susceptibility profile, the infection source, and the antimicrobials received were prospectively recorded.

## Statistical analysis

Patient characteristics were expressed as n (%) for categorical variables and median (interquartile range (IQR)) for continuous variables. Comparisons were made with exact Fisher tests for categorical variables and Wilcoxon tests for continuous variables.

The primary outcome measure was 60-day mortality. We used an IPTW estimator, which is the inverse of the patients' predicted probability of being in the Early-CS group, on the basis of their baseline covariates. The IPTW estimator creates a pseudo-population in which baseline patient differences are balanced between treatment groups. The impact of early CS on 60-day mortality was estimated by a two-step process: 1) weight estimation by the IPTW estimator, and 2) estimation of the impact of early CS on 60-day mortality using a weighted Cox model. Weighted Fine and Gray sub-distribution competing risk models [33] were used to estimate the risk of HAP-VAP and ICU-BSI, considering the competing ICU death and ICU discharge.

As a first step, the weight model, a non-parsimonious multivariable logistic regression model, was constructed to estimate each patient's predicted probability of being in the Early-CS group. We included in the weight model the following covariates: time since symptom onset and ICU admission, time between hospital and ICU admission, age, gender, comorbidities including presence of chronic cardio-vascular, respiratory and kidney chronic diseases, clinical and laboratory features on admission, renal SOFA item, $PaO_2/FiO_2$, lymphocyte, neutrophil, monocyte counts, ferritin, C-reactive protein and D-Dimers, treatments received on admission including Lopinavir-Ritonavir and Tocilizumab. All variables included in the weight model reflected knowledge available at baseline [34–36]. To avoid extreme weights, we used stabilized weights, and to ensure respect of the positivity assumption weights were truncated at the 1-99th percentile [37]. For the second step, we used a weighted Cox proportional-hazard model to estimate the risk of death within the first 60 days of ICU stay of early CS. A hazard ratio >1 indicated an increased risk of death. The proportionality of hazard risk for early CS was tested using martingale residuals. A further analysis using a raw (non-weighted) multivariable Cox proportional-hazard model was performed to confirm the results obtained with the IPTW model. All models were stratified by center. The analyses were carried out similarly for the risks of HAP-VAP and ICU-BSI using subdistribution hazard models with ICU discharge as competing risk instead of Cox models.

Similar analyses were performed for the patients receiving high doses of corticosteroids.

We tested interactions between age, inflammation and mechanical ventilation. Subgroup analyses were then performed among the patients with inflammation or not, older or younger than 60 years old, on mechanical ventilation or not on admission and among the patients admitted before and after day 7 following the first COVID symptoms and their potential subcategories depending on the presence of interactions [38]. In post hoc analyses, we also performed sensitivity analyses among the patients with high and low levels of C-reactive protein, D-Dimers and ferritin, and tested the interactions between these covariates and age. Complete case analyses were also performed for most of the subgroup analyses.

For all tests, a two-sided α of 0.05 was considered as significant. Missing baseline variables were handled by multiple imputation with only one dataset using proc MI with SAS software. All statistical analyses were performed with SAS software, Version 9.4 (SAS Institute, Cary, NC).

## Results

### Database description

From 15 February to 1 May 2020, 355 patients with laboratory confirmed COVID-19 were admitted to ICUs of the OutcomeRea<sup>TM</sup> network. Of these, 303 were included in the study (Fig 1). Overall, 238 (78.8%) were male, with a median age (IQR) of 61 (53–70) years. The sex distribution and median age of included and excluded patients were similar (Table 1). One or more comorbidities were present in 176 patients (58%), with obesity and cardiovascular disease being the most frequently coexisting medical conditions, confirmed in 107 (35.4%) and 79 (26%) patients, respectively. Time from onset of symptoms to ICU admission was 10 (7–12) days, and time from hospital to ICU admission was 2 (1–4) days. On ICU admission, SAPS II was 33 (25–44). 222 (66.6%) patients had moderate to severe ARDS. Overall, 106 (35%) patients received IMV, 38 (12.6%) oxygen by mask or nasal prongs, 113 (37.2%) high-flow nasal cannula (HFNC) therapy and 24 (8%) continuous positive airway pressure (CPAP). The median lymphocyte count on admission was 0.8 G/L [0.5; 1.1], and 202 (66.7%) patients had inflammation on admission. The median follow-up time was 12 days (7–20).

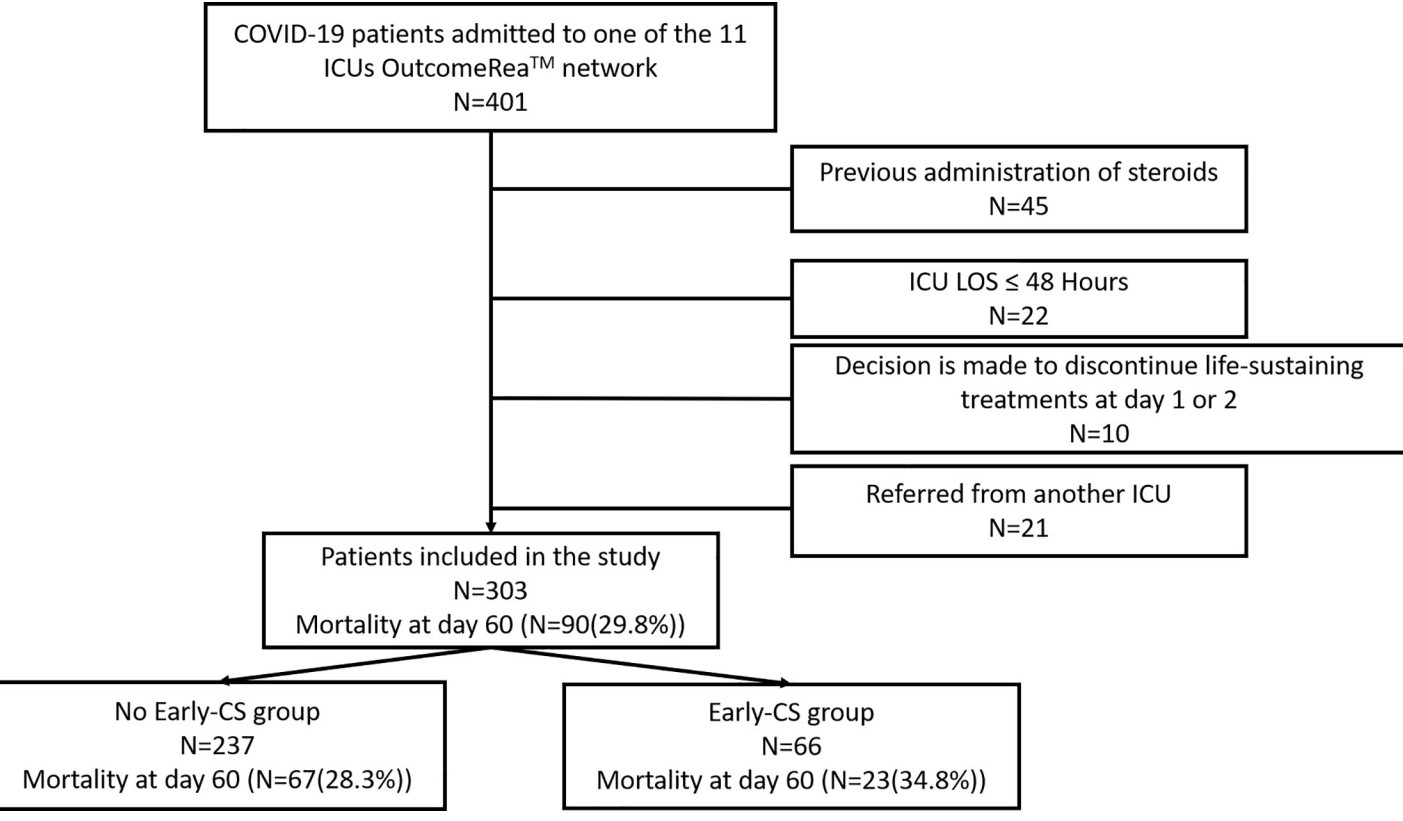

**Fig 1. Flow chart.** ICU: Intensive care unit; LOS: Length of stay; CS: Corticosteroids.

**Table 1. Comparison of the baseline characteristics of patients with and without early corticosteroids after ICU admission.**

| | All | Non-early CS | Early-CS | |
|---|---|---|---|---|
| **Number of patients** | 303 | 237 | 66 | . |
| **Baseline characteristics** | | | | |
| Age | 61 [53; 70] | 61 [53; 70] | 62.5 [55; 71] | 0.42 |
| Gender (Male) | 239 (78.8) | 183 (77.2) | 56 (84.8) | 0.18 |
| Body-mass index, kg/cm$^2$ * (miss = 10) | 28.4 [25.6; 32.2] | 28.6 [25.6; 32.2] | 27.3 [25.1; 32.1] | 0.22 |
| Body-mass index $\geq$ 30 | 107 (35.4) | 84 (35.4) | 23 (34.8) | 0.93 |
| **Comorbidities** | | | | |
| At least one comorbidity | 176 (58) | 130 (54.9) | 46 (69.7) | 0.03 |
| Chronic Liver Failure | 4 (1.4) | 2 (0.8) | 2 (3) | 0.17 |
| Chronic Cardiovascular Disease | 79 (26) | 61 (25.7) | 18 (27.3) | 0.80 |
| Chronic Respiratory Failure | 33 (10.8) | 22 (9.3) | 11 (16.7) | 0.09 |
| Chronic Kidney Disease | 22 (7.2) | 14 (5.9) | 8 (12.1) | 0.09 |
| Immunosuppression§ | 11 (3.6) | 5 (2.1) | 6 (9.1) | <0.01 |
| Time between symptoms and ICU admission | 10 [7; 12] | 10 [7; 12] | 10 [8; 13] | 0.12 |
| **Treatment before admission** | | | | |
| Angiotensin converting enzyme inhibitor | 59 (19.4) | 46 (19.4) | 13 (19.7) | 0.96 |
| Statin | 36 (11.8) | 29 (12.2) | 7 (10.6) | 0.72 |
| Non-steroidal anti-inflammatory drug | 12 (4) | 10 (4.2) | 2 (3) | 0.66 |
| Lopinavir Ritonavir | 5 (1.6) | 3 (1.3) | 2 (3) | 0.32 |
| Choloroquine | 5 (1.6) | 4 (1.7) | 1 (1.5) | 0.92 |
| Immunomodulatory treatment** | 9 (3) | 7 (3) | 2 (3) | 0.97 |
| **Characteristics on admission** | | | | |
| SAPS II score | 33 [25; 44] | 32 [24; 42] | 37 [29; 48] | <0.01 |
| SOFA score | 4 [3; 7] | 4 [2; 6] | 5 [3; 8] | <0.01 |
| SOFA respiratory item (>2) | 171 (56.4) | 122 (51.5) | 49 (74.2) | <0.01 |
| SOFA cardio-vascular item (>2) | 75 (24.8) | 55 (23.2) | 20 (30.3) | 0.24 |
| SOFA Kidney item (>2) | 35 (11.6) | 23 (9.7) | 12 (18.2) | 0.06 |
| Neurologic failure (GCS < 15) | 15 (5) | 11 (4.6) | 4 (6.1) | 0.64 |
| Body temperature > 39°C | 96 (31.6) | 82 (34.6) | 14 (21.2) | 0.04 |
| **Severity of ARDS** | | | | |
| No ARDS PaO2/FiO2 > 300 | 37 (12.2) | 35 (14.8) | 2 (3) | <0.01 |
| Mild: PaO2/FiO2 200–300 | 44 (14.6) | 39 (16.5) | 5 (7.6) | . |
| Moderate: PaO2/FiO2 100–200 | 133 (43.8) | 102 (43) | 31 (47) | . |
| Severe: PaO2/FiO2 < 100 | 89 (29.4) | 61 (25.7) | 28 (42.4) | . |
| **Ventilatory support on admission** | | | | |
| Mechanical ventilation on admission | 106 (35) | 81 (34.2) | 25 (37.9) | 0.58 |
| Non-invasive positive pressure ventilation | 75 (24.8) | 56 (23.6) | 19 (28.8) | 0.39 |
| High flow nasal cannula | 113 (37.2) | 94 (39.7) | 19 (28.8) | 0.11 |
| Continuous positive airway pressure | 24 (8) | 13 (5.5) | 11 (16.7) | <0.01 |
| Oxygen by mask or nasal prongs | 38 (12.6) | 31 (13.1) | 7 (10.6) | 0.59 |
| **Laboratory features on admission** | | | | |
| Leucocytes (miss = 13˚) | 7800 [5750; 10600] | 7600 [5640; 10300] | 8793.1 [7000; 12200] | 0.02 |
| Neutrophils (miss = 39) | 6900 [4700; 10140] | 6800 [4600; 9700] | 7415 [5240; 11600] | 0.06 |
| Lymphocytes (miss = 39) | 800 [530; 1170] | 800 [600; 1100] | 775 [500; 1260] | 0.91 |
| Monocytes (miss = 47) | 350.2 [210; 600] | 367.1 [200; 600] | 350 [240; 600] | 0.91 |
| CRP (miss = 47) | 130.6 [77; 205.4] | 148.8 [83.8; 232] | 162 [82; 224] | 0.80 |
| Ferritin (miss = 103) | 867.6 [490; 1712.6] | 990.6 [547.2; 1801] | 1310.5 [726; 2144.7] | 0.09 |

*(Continued)*

**Table 1.** (Continued)

| | All | Non-early CS | Early-CS | |
|---|---|---|---|---|
| DDimers (miss = 89) | 1155 [700; 2670] | 1900 [902; 5756.6] | 1702.5 [766; 4700] | 0.4 |
| Inflammation* | 202 (66.7) | 159 (67.1) | 43 (65.2) | 0.77 |
| **Steroid characteristics on admission** | | | | |
| Corticosteroids | 66 (21.8) | 0 (0) | 66 (100) | <0.01 |
| High dose of corticosteroids | 55 (18.2) | 0 (0) | 55 (83.3) | <0.01 |
| Low dose of corticosteroids | 11 (3.6) | 0 (0) | 11 (16.7) | <0.01 |
| Dexamethasone | 47 (15.6) | 0 (0) | 47 (71.2) | <0.01 |
| HSHC | 8 (2.6) | 0 (0) | 8 (12.1) | <0.01 |
| Methylprednisolone | 2 (0.6) | 0 (0) | 2 (3) | <0.01 |
| Prednisolone | 9 (3) | 0 (0) | 9 (13.6) | <0.01 |
| First corticosteroids after Day 3 | 94 (31) | 94 (39.7) | 0 (0) | <0.01 |
| High dose of steroids after Day 3 | 87 (28.8) | 87 (36.7) | 0 (0) | <0.01 |
| Low dose of steroids after Day 3 | 7 (2.4) | 7 (3) | 0 (0) | 0.16 |
| **Other treatments on admission** | | | | |
| Lopinavir Ritonavir | 109 (36) | 81 (34.2) | 28 (42.4) | 0.22 |
| Hydroxychloroquine | 33 (10.8) | 21 (8.9) | 12 (18.2) | 0.03 |
| Tocilizumab | 25 (8.2) | 17 (7.2) | 8 (12.1) | 0.20 |
| Anakinra | 22 (7.2) | 0 (0) | 22 (33.3) | <0.01 |
| Preventive anticoagulation | 217 (71.6) | 166 (71.2) | 49 (74.2) | 0.63 |
| Curative anticoagulation | 75 (24.8) | 52 (22.3) | 21 (31.8) | 0.11 |
| Preventive anticoagulation during ICU stay | | 173 (74.2) | 55 (83.3) | 0.13 |
| Curative anticoagulation during ICU stay | | 111 (47.6) | 36 (54.5) | 0.32 |
| **LOS and Mortality** | | | | |
| ICU LOS | 12 [7; 20] | 12 [6; 20] | 11 [7; 19] | 0.91 |
| ICU Death | 86 (28.4) | 64 (27) | 22 (33.3) | 0.31 |
| Death at day 60 | 90 (29.8) | 67 (28.3) | 23 (34.8) | 0.30 |
| **Adverse events due to corticosteroids** | | | | |
| Hyperglycemia | 105 (34.6) | 59 (24.9) | 46 (69.7) | <0.01 |
| Mean daily dose of insulin | 9 [0; 44.2] | 6.2 [0; 37.1] | 27.2 [4.4; 58.4] | <0.01 |
| Number of days under MV | 7 [0; 16] | 8 [0; 16] | 5 [0; 14] | 0.35 |
| VFD | 4 [1; 7] | 3 [1; 7] | 4 [1; 8] | 0.12 |
| ICU-BSI | 43 (14.2) | 30 (12.7) | 13 (19.7) | 0.15 |
| HAP-VAP | 95 (31.4) | 70 (29.5) | 25 (37.9) | 0.20 |
| VAP | 90 (29.8) | 67 (28.3) | 23 (34.8) | 0.30 |

Inflammation* At least 2 of the following criteria: a Ferritin > 1000 μg/l or D-Dimers > 1000 μg/l or C-Reactive Protein > 100 mg/dL VFD: Ventilatory free days; BSI: Blood stream infection, HAP-VAP: hospital-acquired pneumonia and ventilator-associated pneumonia. LOS: Length of stay; HSHC: Hydrocortisone hemisuccinate ICU: intensive care unit.

**: Steroids or Tocilizumab or Anakinra.

## Early- versus non-early corticosteroids group

Sixty-six patients were in the Early-CS group including 47 patients receiving dexamethasone. In the Non-early-CS group, 94/237(39.7%) patients received corticosteroids during their ICU stay (S1 Fig). Lopinavir-ritonavir was administered in 109 (36%) patients, hydroxychloroquine in 33 (10.8%), tocilizumab in 25 (8.2%), and anakinra in 22 (7.2%). The comparison of baseline characteristics between the Early- and the Non-early-CS groups is shown in Table 1. Both groups had similar ICU ventilatory-free days, ICU-LOS and 60-day mortality. The variables

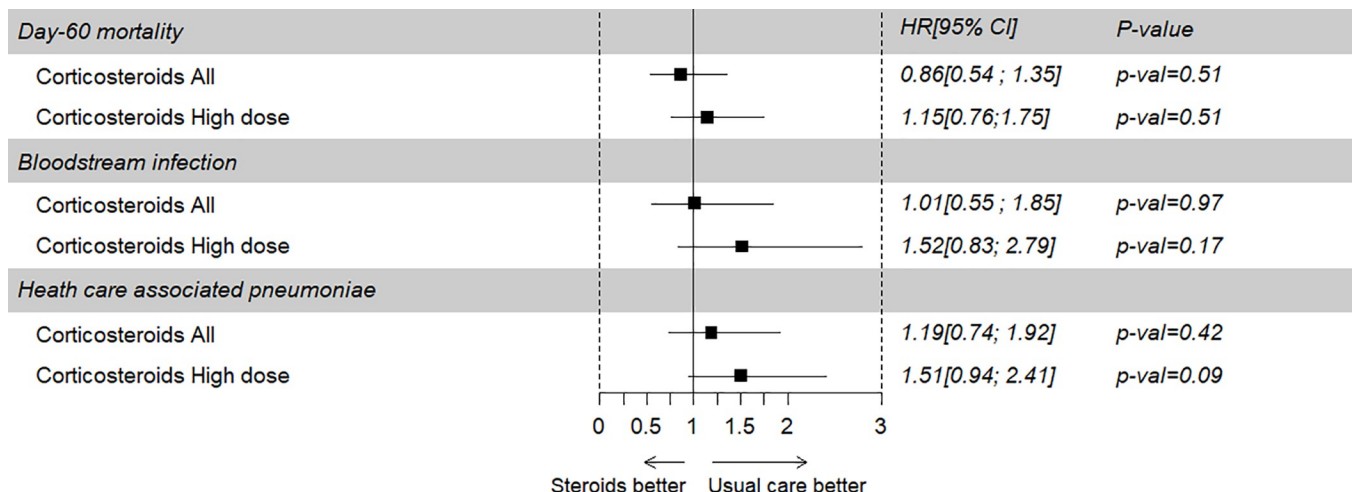

**Fig 2. Effect of corticosteroids on ICU death and the occurrence of blood stream infection and HAP-VAP of patients in the main cohort.** HAP-VAP: hospital-acquired pneumonia and ventilator-associated pneumonia; HR: Hazard ratio; CI: Confidence interval.

used to determine the risk for early CS administration and validity of the model are shown in Fig 2 and S2–S4 Figs.

## Primary endpoint

Overall, 60-day mortality was 29.8% with no difference between the Early-CS subgroup and the Non-early-CS subgroup (34.8% versus 28.3%, p = 0.28). After weighted Cox model analysis, the risk of death at day 60 was similar in patients with and without early CS (HRw = 0.86, CI 95% 0.54 to 1.35, p = 0.51; Fig 2). Similar results were observed in a sensitivity analysis using truncated weights (S1 Table). In a multivariable Cox survival model without weighting on IPTW, early CS therapy was not associated with the risk of mortality (S2 Table). Similar results were observed when limiting the analysis to patients of the Early-CS group receiving high doses of CS ($_{IPTW}$HR = 1.15, CI 95%, 0.76 to 1.75, p = 0.51, Fig 2).

## Subgroup analyses and secondary endpoints

Subgroup analyses showed that early CS administration was associated with a lower mortality rate in patients aged 60 years or more ($_{IPTW}$HR, 0.53; 95% CI, 0.3–0.93; p = 0.03). An interaction was found between age and inflammation. As a result, subgroup analyses were also performed among older and younger patients, with and without inflammation.

CS therapy was associated with higher mortality in patients younger than 60 without inflammation on admission ($_{IPTW}$HR, 5.01; 95% CI, 1.05–23.88; p = 0.04) (Fig 3 and S3 Table). Results tended to be similar for the patients with high or low levels of D-Dimers, C-reactive protein and ferritin and in complete case analyses (S5 and S6 Figs and S4 and S5 Tables).

## Safety and healthcare associated infections

The main adverse events recorded are given in Table 1. The main differences observed between patients who received early CS or not were a higher rate of developing at least one hyperglycemia event: 46 (69.7%) vs. 59 (24.9%), (p < 0.01), and a higher median daily dose of insulin: 27.2 Ui [4.4; 58.4] vs. 6.2 Ui [0; 37.1], (p< 0.01), respectively. There were no differences between patients who received early CS or not in the rate of developing at least one

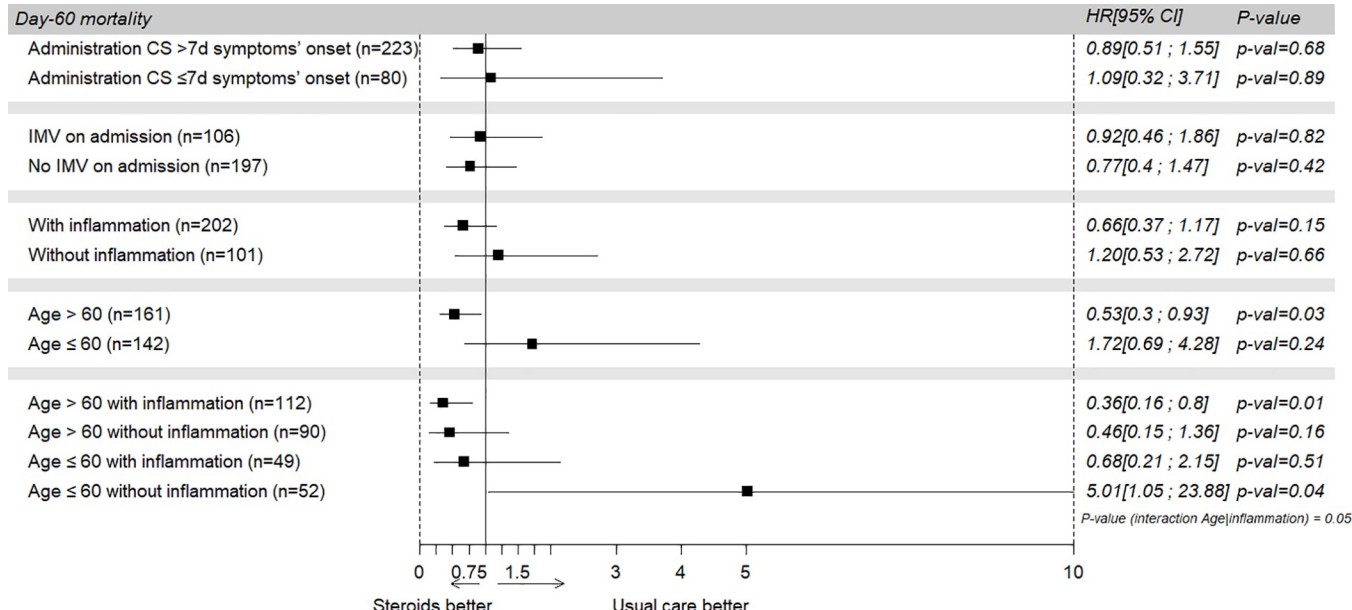

**Fig 3. Effect of corticosteroids on ICU death in different subgroups.** CS: Corticosteroids; IMV: Invasive mechanical ventilation; HR: Hazard Ratio; CI: Confidence Interval.

episode of ICU-BSI: 13 (19.7%) vs. 30 (19.7%) (p = 0.15), and at least one episode of HAP-VAP: 25 (37.9%) vs. 70 (29.5%) (p = 0.20), respectively.

After weighted Fine & Gray subdistribution survival model analyses, the risk of ICU-BSI and the risk of HAP-VAP at day 60 did not differ between patients with and without early CS therapy (SubHRw = 1.01 for ICU-BSI, CI 95% 0.55 to 1.85, p = 0.97 and SubHRw = 1.19 for HAP-VAP, CI 95% 0.74 to 1.92, p = 0.42, respectively) (Fig 2 and S6 and S7 Tables).

## Discussion

We report a multicenter observational study involving 11 French ICUs in the Outcomerea© network that assessed the efficacy and safety of early CS therapy in patients admitted to the ICU for COVID-19 pneumonia.

Early administration of CS, during the first two days after ICU admission, was not significantly associated with 60-day mortality. Similar results were observed in patients who received high doses of corticosteroids. Importantly, early CS administration was beneficial in older ICU patients but not in patients younger than 60. Early CS seemed to be potentially disadvantageous in younger patients without inflammation on admission. It was also associated with a significant increase in the risk of hyperglycemia and insulin requirement and did not affect significantly the incidence of HAP-VAP and/or ICU-BSI.

The benefit in patients older than 65 had already been suggested by a subgroup analysis of the MetCOVID trial [15]. One explanation given by the authors was that older patients had more inflammation on admission. In our cohort, CRP and ferritin levels were similar in the two age groups. However, older patients had a lower lymphocyte count, a higher DDimer level and a higher severity score (S3 Table). Another hypothesis could be therefore that the more severe the disease, the more effective are steroids. These results need confirmation, since the Recovery trial found a protective effect of steroids for patients younger than 70 years.

Our results suggested that steroids should be avoided in younger patients in the absence of inflammation. Such results could also be related to the severity of the pneumonia. Indeed, in

our cohort, patients without inflammation had less severe pneumonia symptoms (S5 Table). Our results are consistent with those of other studies which also found that steroids were beneficial in the subgroup of patients with inflammation [15, 19, 39]. Another observational study reported similar results but also a deleterious effect of CS for patients with a C-Reactive protein below 10 mg/dL [40]. Furthermore, in the Recovery trial and another meta-analysis, the less severe patients, i.e. those without oxygen, did not benefit from CS [11, 41]. The beneficial effects of steroids could be explained by their potential role in suppressing inflammatory storms, reducing inflammatory exudation, and preventing multiple organ injuries [3, 4, 42]. However, further studies of COVID-19 ARDS are needed to better understand the direct effect of steroids on this particular immune response [43], which is different from that of other bacterial sepsis [44, 45]. In contrast, steroids could have a deleterious effect in the absence of inflammation because they induce immunosuppression [46].

One of the potential consequences of immunosuppression is delayed SARS-CoV-2 RNA clearance [47], which had already been observed in SARS and MERS [6] but not in SARS-COV 2 patients [48]. In addition, the immunosuppression induced by steroids could also lead to a higher risk of superinfections [23, 45]. Several studies have already reported a high rate of ICU-acquired pneumonia in mechanically ventilated COVID-19 patients [49].

There are several reasons why our results are at variance with those of the Recovery trial, which support the use of corticosteroids to reduce death rates. First, our patients received high doses of steroids (20 mg of dexamethasone for 5 days and then 10 mg of dexamethasone for 5 days), which could have been more harmful than lower doses. To date, only a few studies have assessed high doses of steroids. One observational study reported that a higher dose was associated with harmful effects [50]. Only the CoDEX trial has assessed dexamethasone at a higher dose (up to 20 mg per day), reporting a positive effect measured as a composite of days alive and free of mechanical ventilation [13]. However, in the CoDEX trial, 28-day mortality was not different between high doses of steroids and placebo. Results from other clinical trials are pending before definitive conclusions can be drawn. Second, our patients received steroids from symptom onset, later than in the Recovery trials (10 days versus 8 days), which could have been too late to prevent or reverse the damage caused by extensive inflammation. Third, one third of the patients in the Non-early-CS group finally received steroids, which could have minimized differences between the two groups. Fourth, the absence of benefit of CS in our study could be related to the significant increase in the rate of hyperglycemia and in insulin use. Indeed, the absence of glycemic control in critically ill patients is associated with a demonstrated increased risk of death [51]. Finally, other immunomodulatory treatments could have interfered with the effects of steroids. In our cohort, some patients also received tocilizumab or Anakinra. Such treatments are under evaluation and preliminary results of interleukin-6 or interleukin-1 blockade and/or anti-TNF are varying. Most studies of tocilizumab were gathered in a meta-analysis which showed that it did not reduce short-term mortality [52, 53]. Anakinra could reduce the risk of invasive mechanical ventilation and death [54] but results from randomized controlled trials are still pending.

Hypoxemia in COVID-19 patients is also secondary to hypercoagulability [55, 56] which is the cause of arterial or venous thrombotic events including microcirculation thrombosis [57] and pulmonary embolism [58]. Elevated D-Dimer levels are associated with this increased risk of pulmonary embolism [59]. Anticoagulation strategies are consequently of paramount importance in COVID-19 patients [60]. The benefit of steroids might therefore be confounded by hypercoagulability and anticoagulation. In our cohort, anticoagulation strategies were quickly adapted to recommendations. To explore the impact of hypercoagulability, we performed subgroup analyses with patients with various D-Dimer levels and found that patients with high D-Dimer levels and older age benefitted from steroids.

Finally, in our cohort, mostly admitted during the first wave of the pandemic in France, very few patients received before ICU admission anticoagulation, anti-viral or immunomodulatory treatments, which might have worsened the observed outcomes [61].

Our results might therefore be different now that more patients receive early treatment including steroids and anticoagulation, which could avoid hospitalization or admission to the ICU [62].

The strength of our study resides in our subgroup analyses, the 60-day endpoint and the use of weighted models that minimize the weight of patients unlikely to have received corticosteroids. We also excluded all patients previously exposed to steroids to reduce immortality time bias. Our study has several limitations. First, despite the use of propensity score analyses to draw causal inferences the study was observational, and potential unmeasured confounders may still have biased our results. Hence, we acknowledge that variables might not have included treatments received before admission and antiplatelet agents. Second, our study dealt with heterogeneity in the prescription of corticosteroids in terms of drugs, doses, and duration but also in terms of other immunomodulatory treatments. In addition, some of the patients in the Non-early-CS group received finally corticosteroids. We also considered only the first episode of BSI or HAP-VAP. Finally, there was a substantial number of missing data concerning certain laboratory features. For this reason, we performed several sensitivity analyses.

## Conclusion

In conclusion, we were unable to identify a beneficial effect of steroids on 60-day mortality in critically ill COVID-19 pneumonia patients, mostly because of variations in the clinical characteristics of the patients and in the choice, dose and duration of steroids. However, we showed that inflammation and age could be important criteria to determine which patients would benefit the most from early steroid therapy. This finding should be confirmed prospectively. Personalization of the administration of steroids and other immunomodulatory treatments based on biomarkers warrants further investigation.

## Supporting information

**S1 Fig. Distribution of the start of steroids in the population studied.**
(DOCX)

**S2 Fig. Propensity score: Multivariate regression logistic analysis for the factors associated with receiving early corticosteroids.** ICU: Intensive care unit; SOFA: Sequential Organ Failure Assessment; AUC: 0.76.
(DOCX)

**S3 Fig. Histograms of propensity score by treatment group: Early steroids and non-early corticosteroids.** * a Ferritin > 1000 µg/l or D-Dimers > 1000 µg/l or C-Reactive Protein > 100 mg/dL.
(DOCX)

**S4 Fig.** Standardized differences in the covariates included in the propensity score, before (in blue) and after (in red) weighting.
(DOCX)

**S5 Fig. Subgroup analyses based on C-Reactive protein, ferritin, and DDimer levels.**
(DOCX)

**S6 Fig. Subgroup analyses based on C-Reactive protein, ferritin, and DDimer levels and interaction with age.**
(DOCX)

**S1 Table Cox model for 60-day mortality with ponderation on IPTW for the impact of early corticosteroids IPTW: Inverse Probability of treatment weight; HR: Hazard Ratio.**
(DOCX)

**S2 Table. Multivariate survival analyses of the factors associated with 60-day mortality.** HR: Hazard Ratio; SOFA: Sequential Organ Failure Assessment; HR: Hazard Ration.
(DOCX)

**S3 Table. Comparison between older and younger patients and between patients with and without inflammation.** Inflammation* Ferritin >1000 µg/l or D-Dimers >1000 µg/l or C-Reactive Protein >100 mg/dL; VFD: Ventilatory free days; BSI: Blood stream infection, HAP-VAP: hospital-acquired pneumonia and ventilator-associated pneumonia. LOS: Length of stay; HSHC: Hydrocortisone hemisuccinate ICU: intensive care unit; SOFA: Sequential organ Failure assessment; SAPS II: simplified acute physiology score.
(DOCX)

**S4 Table. Occurrence of HAP-VAP: Competing risk models with death or leaving alive as competing risks.** HAP-VAP: Ventilator associated pneumoniae; SubHR: Sub hazard ratio.
(DOCX)

**S5 Table. Occurrence of ICU-BSI: Competing risk models with death or leaving alive as competing risks.** ICU BSI: Intensive care unit blood stream infection; SubHR: Sub Hazard Ratio.
(DOCX)

**S1 Dataset.**
(XLSX)

**S2 Dataset. Glossary dataset.**
(XLSX)

## Acknowledgments

The OutcomeRea^TM study group thank Jeffrey Watts for advice on the English version of the manuscript.

OUTCOMEREA Study Group: Jean-François Timsit (E mail: jean-francois.timsit@aphp.fr), Elie Azoulay, Maïté Garrouste-Orgeas, Jean-Ralph Zahar, Bruno Mourvillier, Michael Darmon, Christophe Clec'h, Corinne Alberti, Stephane Ruckly, Sébastien Bailly, Aurélien Vannieuwenhuyze, Romain Hernu, Christophe Adrie, Carole Agasse, Bernard Allaouchiche, Olivier Andremont, Pascal Andreu, Laurent Argaud, Claire Ara-Somohano, Elie Azoulay, Francois Barbier, Déborah Boyer, Jean-Pierre Bedos, Thomas Baudry, Jérome Bedel, Julien Bohé, Lila Bouadma, Jeremy Bourenne, Noel Brule, Cédric Brétonnière, Charles Cerf, Frank Chemouni, Christine Cheval, Julien Carvelli, Elisabeth Coupez, Martin Cour, Claire Dupuis, Etienne de Montmollin, Loa Dopeux, Anne-Sylvie Dumenil, Jean-Marc Forel, Marc Gainnier, Charlotte Garret, Dany Goldgran-Tonedano, Steven Grangé, Antoine Gros, Hédia Hammed, Akim Haouache, Romain Hernu, Tarik Hissem, Vivien Hong Tuan Ha, Sébastien Jochmans, Jean-Baptiste Joffredo, Hatem Kallel, Guillaume Lacave, Virgine Laurent, Alexandre Lautrette, Clément Le bihan Eric Magalhaes, Virgine Lemiale, Guillaume Marcotte, Jordane Lebut, Maxime Lugosi, Sibylle Merceron, Benoît Misset, Mathild Neuville, Laurent

Nicolet, Johanna Oziel, Laurent Papazian, Juliette Patrier, Benjamin Planquette, Aguila Radjou, Marie Simon, Romain Sonneville, Jean Reignier, Bertrand Souweine, Carole Schwebel, Shidasp Siami, Romain Sonneville, Nicolas Terzi, Gilles Troché, Marie Thuong, Guillaume Thierry, Marion Venot, Sondes Yaacoubi, Olivier Zambon, Julien Fournier, Stéphanie Bagur, Mireille Adda, Vanessa Vindrieux, Sylvie de la Salle, Pauline Enguerrand, Vincent Gobert, Stéphane Guessens, Helene Merle, Nadira Kaddour, Boris Berthe, Samir Bekkhouche, Kaouttar Mellouk, Mélaine Lebrazic, Carole Ouisse, Diane Maugars, Christelle Aparicio, Igor Theodose, Manal Nouacer, Veronique Deiler, Fariza Lamara, Myriam Moussa, Atika Mouaci and Nassima Viguier.

## Author Contributions

**Conceptualization:** Claire Dupuis, Jean-François Timsit.

**Data curation:** Claire Dupuis, Dany Goldgran-Toledano, Jean Reignier, Carole Schwebel, Julien Domitile, Mathilde Neuville, Shidasp Siami, Stéphane Ruckly, Bruno Mourvillier, Virginie Laurent, Marc Gainnier, Jean-François Timsit.

**Formal analysis:** Claire Dupuis, Stéphane Ruckly.

**Methodology:** Claire Dupuis, Niccolò Buetti, Stéphane Ruckly, Corinne Alberti, Sebastien Bailly, Bertrand Souweine, Jean-François Timsit.

**Project administration:** Jean-François Timsit.

**Resources:** Dany Goldgran-Toledano, Jean Reignier, Carole Schwebel, Mathilde Neuville, Shidasp Siami, Stéphane Ruckly, Bruno Mourvillier, Virginie Laurent, Marc Gainnier, Jean-François Timsit.

**Supervision:** Stéphane Ruckly, Corinne Alberti, Sebastien Bailly, Jean-François Timsit.

**Validation:** Niccolò Buetti, Jean Reignier, Carole Schwebel, Mathilde Neuville, Moreno Ursino, Shidasp Siami, Stéphane Ruckly, Corinne Alberti, Bruno Mourvillier, Sebastien Bailly, Virginie Laurent, Marc Gainnier, Bertrand Souweine, Jean-François Timsit.

**Writing – original draft:** Claire Dupuis, Bertrand Souweine, Jean-François Timsit.

**Writing – review & editing:** Etienne de Montmollin, Niccolò Buetti, Bertrand Souweine, Jean-François Timsit.

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
