## [Decision Letter · Decision Letter 0]

8 Jun 2021

PONE-D-21-11495

Impact of Early Corticosteroids on 60-day Mortality in Critically Ill Patients with COVID-19: A Multicenter Cohort Study of the OUTCOMEREA Network

PLOS ONE

Dear Dr. Dupuis,

Thank you for submitting your manuscript to PLOS ONE. After careful consideration, we feel that it has merit but does not fully meet PLOS ONE’s publication criteria as it currently stands. Therefore, we invite you to submit a revised version of the manuscript that addresses the points raised during the review process.

We look forward to receiving your revised manuscript.

Kind regards,

Aleksandar R. Zivkovic

Academic Editor

PLOS ONE

2. Thank you for including your ethics statement:  "In accordance with French law, the OutcomeReaTM database has been approved by the French Advisory Committee for Data Processing in Health Research (CCTIRS) and the French Informatics and Liberty Commission (CNIL, registration no. 8999262). The database protocol was submitted to the Institutional Review Board of the Clermont-Ferrand University Hospital (Clermont-Ferrand, France), who waived the need for informed consent (IRB no. 5891).   

a. Please amend your current ethics statement to state whether the IRB specifically reviewed and approved your study.

3. Please include your tables as part of your main manuscript and remove the individual files. Please note that supplementary tables (should remain/ be uploaded) as separate "supporting information" files

"JFT declares no conflict of interest related to the submitted work. Outside the submitted work, JFT declares participation in advisory boards for Merck, Pfizer, Gilead, Nabriva and Paratek, lecture fees from Biomerieux, Pfizer and Merck, and research grants to his research unit from Merck, 3M, Astelas and Thermofisher.

Dr. Buetti is currently receiving a mobility grant from the Swiss National Science Foundation (grant number: P400PM_183865) and a grant from the Bangerter-Rhyner Foundation. These grants support his fellowship in Paris."

We note that one or more of the authors are employed by a commercial company: ICUREsearch

5.We note that you have indicated that data from this study are available upon request. PLOS only allows data to be available upon request if there are legal or ethical restrictions on sharing data publicly. For information on unacceptable data access restrictions, please see http://journals.plos.org/plosone/s/data-availability#loc-unacceptable-data-access-restrictions.

6. One of the noted authors is a group or consortium [The OutcomereaTM study group - outcomerea.fr]. In addition to naming the author group, please list the individual authors and affiliations within this group in the acknowledgments section of your manuscript. Please also indicate clearly a lead author for this group along with a contact email address.

8. Your ethics statement should only appear in the Methods section of your manuscript. If your ethics statement is written in any section besides the Methods, please delete it from any other section.

**Comments to the Author**

Reviewer #1: This is a retrospective cohort study where the authors conclude that overall early(<48h) or non early CS (?>48h) steroid use has no statistically significant effect on 60 day mortality. In a subgroup analysis, patients who were >60 years of age may have favorable outcomes and those <60 years and with "low inflammation" seem to be at risk of harm. Low inflammation is defined as cut offs for CRP/Ddimer and Ferritin.

Major concern

Heterogeneity of treatment effect with steroids is well known. In a recently published observational analysis by Keller et al.. Hosp. Med 2020;8;489-493 in patients with CRP ≥20mg/dL, corticosteroid treatment was associated with a 75-80% reduction in the composite severe outcomes of mechanical ventilation and mortality (odds ratio [OR], 0.23; 95% CI, 0.08-0.70). Conversely, and as importantly, among those with CRP <10mg/dL, there was increase in severe outcomes (OR, 2.64; 95% CI, 1.39-5.03). Within the CRP category of 10-19.9mg/dL, corticosteroids were neither beneficial nor harmful (OR, 1.05; 95% CI, 0.46-2.39).

Since the authors noted significant interaction between treatment and initial d dimer/ CRP level was recommend to do a post hoc adjusted analysis within each of the predefined subgroup variables

Definition/Group assignment

non early CS group is not defined. It appears it would include all patients who got steroids >48 after ICU admission but that does not seem to be the case since only 96/236 patients ended up getting steroids (Table 1). Also this is 40% and not 31.1% as mentioned in line 168. All 66 patients in the early CS group got steroids.

Missing values

Nearly 30% of CRP values are missing and nearly 43% of ferritin and d dimer are missing. With these large datasets missing multiple imputation might not be sufficient. Perhaps the missing values can be excluded for the analysis in which case the findings would be considered hypothesis generating.

How were they dealt with in the model?

Minor concerns

Figure 1- the consort diagram numbers do not add up. I suspect some of the exclusion categories had shared patients

Figure 3- numbers do not add up, adding age<60+age >60 =303; n was 302!

Reviewer #2: Impact of Early Corticosteroids on 60-day Mortality in Critically Ill Patients with COVID-19: A Multicenter Cohort Study of the OUTCOMEREA Network

Summary

This is a report of the associated mortality reduction of late-stage steroids in COVID-19 hospitalization. In general, the conclusions are supported by the data.

Major Comments

Table 1. Please list the COVID-19 specific treatment given before hospitalization. If this is uknown, please list as a limitation.

Table 1. Please list the antiplatelet agents and antithrombotics used in the hospital.

Discussion

Please discuss and cite the data supporting hemagglutination and microthrombosis as a determinant of hypoxemia and “COVID-19” and how the antiplatelet and antithrombotic drugs used in the study could have influenced the results.

Please insert a paragraph concerning the prehospital treatment received. If little or no prehospital therapy is received then this worked to worsened the outcomes observed. Please cite and mention this paper concerning significant reductions in events with prehospital therapy: Procter, MD, B. C., APRN, FNP-C, C. R. M., PA-C, MPAS, V. P., PA-C, MPAS, E. S., PA-C, MPAS, C. H., & McCullough, MD, MPH, P. A. (2021). Early Ambulatory Multidrug Therapy Reduces Hospitalization and Death in High-Risk Patients with SARS-CoV-2 (COVID-19). International Journal of Innovative Research in Medical Science, 6(03), 219 - 221. https://doi.org/10.23958/ijirms/vol06-i03/1100

Please indicate how the present results will be applicable now that more patients receive early treatment and avoid the hospital altogether, see and cite: McCullough PA, Alexander PE, Armstrong R, Arvinte C, Bain AF, Bartlett RP, Berkowitz RL, Berry AC, Borody TJ, Brewer JH, Brufsky AM, Clarke T, Derwand R, Eck A, Eck J, Eisner RA, Fareed GC, Farella A, Fonseca SNS, Geyer CE Jr, Gonnering RS, Graves KE, Gross KBV, Hazan S, Held KS, Hight HT, Immanuel S, Jacobs MM, Ladapo JA, Lee LH, Littell J, Lozano I, Mangat HS, Marble B, McKinnon JE, Merritt LD, Orient JM, Oskoui R, Pompan DC, Procter BC, Prodromos C, Rajter JC, Rajter JJ, Ram CVS, Rios SS, Risch HA, Robb MJA, Rutherford M, Scholz M, Singleton MM, Tumlin JA, Tyson BM, Urso RG, Victory K, Vliet EL, Wax CM, Wolkoff AG, Wooll V, Zelenko V. Multifaceted highly targeted sequential multidrug treatment of early ambulatory high-risk SARS-CoV-2 infection (COVID-19). Rev Cardiovasc Med. 2020 Dec 30;21(4):517-530. doi: 10.31083/j.rcm.2020.04.264. PMID: 33387997.

Reviewer #3: The article is very interesting.

In my opinion, that the cut-off point of ferritin (1000) to define the inflammatory state is very high. It would be convenient a ROC curve with ferritin to identify patients who do not respond.

---

## [Author Response · Author response to Decision Letter 0]

16 Jul 2021

Reviewer #1: This is a retrospective cohort study where the authors conclude that overall early(<48h) or non early CS (?>48h) steroid use has no statistically significant effect on 60 day mortality. In a subgroup analysis, patients who were >60 years of age may have favorable outcomes and those <60 years and with "low inflammation" seem to be at risk of harm. Low inflammation is defined as cut offs for CRP/Ddimer and Ferritin.

Major concern

Heterogeneity of treatment effect with steroids is well known. In a recently published observational analysis by Keller et al.. Hosp. Med 2020;8;489-493 in patients with CRP ≥20mg/dL, corticosteroid treatment was associated with a 75-80% reduction in the composite severe outcomes of mechanical ventilation and mortality (odds ratio [OR], 0.23; 95% CI, 0.08-0.70). Conversely, and as importantly, among those with CRP <10mg/dL, there was increase in severe outcomes (OR, 2.64; 95% CI, 1.39-5.03). Within the CRP category of 10-19.9mg/dL, corticosteroids were neither beneficial nor harmful (OR, 1.05; 95% CI, 0.46-2.39).

Since the authors noted significant interaction between treatment and initial d dimer/ CRP level was recommend to do a post hoc adjusted analysis within each of the predefined subgroup variables 

Thank you for this remark. We have now added subgroup analyses in the supplementary data depending on the levels of CRP, D-Dimers and Ferritin and studied the interaction between age and these covariates. 

Definition/Group assignment

non early CS group is not defined. It appears it would include all patients who got steroids >48 after ICU admission but that does not seem to be the case since only 96/236 patients ended up getting steroids (Table 1). Also this is 40% and not 31.1% as mentioned in line 168. All 66 patients in the early CS group got steroids.

Thank you for this remark. The Non-early-CS subgroup included all the patients who did not receive steroids during the first two days after ICU admission (N=237). Of these, only 94/237 (39.7%), equivalent to 31% (94/303) of the whole cohort, received steroids during their ICU stay but only on day 3 or thereafter. We have modified this result accordingly in the manuscript.

Missing values

Nearly 30% of CRP values are missing and nearly 43% of ferritin and d dimer are missing. With these large datasets missing multiple imputation might not be sufficient. Perhaps the missing values can be excluded for the analysis in which case the findings would be considered hypothesis generating.

How were they dealt with in the model?

Thank you for this truly relevant remark. We acknowledge this limitation. To minimize the effect of missing values we first tried to recover more data. We finally retrieved 47 (15.5%) missing CRP values, 89 (29%) missing D-Dimer values and 103 (34%) missing ferritin values. 

After multiple imputation, 202 (66.7%) of the patients are now considered as having inflammation according to our previous definition.

To consolidate our results, sensitivity analyses were performed. First, we tested the impact of steroids in the subgroups defined only by CRP, D-Dimer and Ferritin levels, respectively. 

Finally, we performed complete case analyses for all those subgroups. The results are given in the supplementary data. 

Minor concerns

Figure 1- the consort diagram numbers do not add up. I suspect some of the exclusion categories had shared patients.

Thank you for this remark. We have modified the flow chart accordingly to avoid any confusion

Figure 3- numbers do not add up, adding age<60+age >60 =303; n was 302!

 Thank you for this remark. We have corrected the mistake in the new version of the manuscript. 

Reviewer #2: Impact of Early Corticosteroids on 60-day Mortality in Critically Ill Patients with COVID-19: A Multicenter Cohort Study of the OUTCOMEREA Network

Summary

This is a report of the associated mortality reduction of late-stage steroids in COVID-19 hospitalization. In general, the conclusions are supported by the data.

Major Comments

Table 1. Please list the COVID-19 specific treatment given before hospitalization. If this is unknown, please list as a limitation.

Thank you for this remark. In table 1 we have now listed all the treatments received by the patients before admission to the ICU prospectively collected in our database. We also acknowledge in the limitations that some other treatments might have been missed. 

Table 1. Please list the antiplatelet agents and antithrombotics used in the hospital.

Thank you for this remark. In Table 1, we have now reported the number of patients under preventive or curative anticoagulation on admission. Unfortunately, we did not record the antiplatelet agents. A sentence has been added in the limitations to underline this missing variable. We have also included these two variables (preventive and curative anticoagulation) in the propensity score.

Discussion

Please discuss and cite the data supporting hemagglutination and microthrombosis as a determinant of hypoxemia and “COVID-19” and how the antiplatelet and antithrombotic drugs used in the study could have influenced the results.

Thank you for this advice. We had a paragraph supporting hemagglutination and microthrombosis as determinants of hypoxemia and COVID-19. 

We also now take into account these potential confounding factors in our analyses by adding anticoagulation and D-Dimer levels in our propensity score and by performing subgroup analyses according to D-Dimer levels. 

“Furthermore, hypoxemia in COVID-19 patients is also secondary to hypercoagulability (Piazza, JAMA, 2020; Huang, Lancet 2020) which is the cause of arterial or venous thrombotic events including microcirculation thrombosis (Ackermann, NEJM, 2020) and pulmonary embolism (Klok, Thromb Res , 2020). Elevated D-Dimer levels are associated with this increased risk of pulmonary embolism (Li, Br J Haematol, 2020). Anticoagulation strategies are consequently of paramount importance in COVID-19 patients (Susen, CC, 2020). The benefit of steroids might therefore be confounded by hypercoagulability and anticoagulation. In our cohort, anticoagulation strategies were quickly adapted to recommendations. To explore the impact of hypercoagulability, we report sub group analyses defined by D-Dimers levels.” 

Please insert a paragraph concerning the prehospital treatment received. If little or no prehospital therapy is received, then this worked to worsened the outcomes observed. Please cite and mention this paper concerning significant reductions in events with prehospital therapy: Procter, MD, B. C., APRN, FNP-C, C. R. M., PA-C, MPAS, V. P., PA-C, MPAS, E. S., PA-C, MPAS, C. H., & McCullough, MD, MPH, P. A. (2021). Early Ambulatory Multidrug Therapy Reduces Hospitalization and Death in High-Risk Patients with SARS-CoV-2 (COVID-19). International Journal of Innovative Research in Medical Science, 6(03), 219 - 221. https://doi.org/10.23958/ijirms/vol06-i03/1100

Please indicate how the present results will be applicable now that more patients receive early treatment and avoid the hospital altogether, see and cite: McCullough PA, Alexander PE, Armstrong R, Arvinte C, Bain AF, Bartlett RP, Berkowitz RL, Berry AC, Borody TJ, Brewer JH, Brufsky AM, Clarke T, Derwand R, Eck A, Eck J, Eisner RA, Fareed GC, Farella A, Fonseca SNS, Geyer CE Jr, Gonnering RS, Graves KE, Gross KBV, Hazan S, Held KS, Hight HT, Immanuel S, Jacobs MM, Ladapo JA, Lee LH, Littell J, Lozano I, Mangat HS, Marble B, McKinnon JE, Merritt LD, Orient JM, Oskoui R, Pompan DC, Procter BC, Prodromos C, Rajter JC, Rajter JJ, Ram CVS, Rios SS, Risch HA, Robb MJA, Rutherford M, Scholz M, Singleton MM, Tumlin JA, Tyson BM, Urso RG, Victory K, Vliet EL, Wax CM, Wolkoff AG, Wooll V, Zelenko V. Multifaceted highly targeted sequential multidrug treatment of early ambulatory high-risk SARS-CoV-2 infection (COVID-19). Rev Cardiovasc Med. 2020 Dec 30;21(4):517-530. doi: 10.31083/j.rcm.2020.04.264. PMID: 33387997.

Thank you for this advice. We had a sentence on previous treatments received by the patients and the recommended citations. 

“Finally, in our cohort, mostly achieved during the first wave of the pandemia in France, very few patients received before ICU admission anticoagulation, anti-viral or immunomodulatory treatments, which might have worsened the observed outcomes (Procter, Research in Medical Science, 2020).

Our results might therefore be different now that more patients receive early treatment including steroids and anticoagulation and therefore prevent from admission hospitalization or admission in ICU (Alexander, Rev Cardiovasc Med. 2020)." 

Reviewer #3: The article is very interesting.

In my opinion, that the cut-off point of ferritin (1000) to define the inflammatory state is very high. It would be convenient a ROC curve with ferritin to identify patients who do not respond.

 Thank you for the remark. We have now included the corresponding ROC curve and a cut-off value of 1816. We decided to keep the same cut-off as proposed by Rubio et al (Rubio-Rivas, International journal of infectious diseases, 2020).

 threshold specificity sensitivity accuracy

Ferritin 0.57 [ 0.5 - 0.64 ] 1816 0.75 0.41 0.65

C REactive Protein 0.55 [ 0.47 - 0.62 ] 187 0.68 0.42 0.6

D-Dimers 0.54 [ 0.47 - 0.61 ] 1099.5 0.7 0.42 0.62

---

## [Editor Report · Decision Letter 1]

21 Jul 2021

Impact of Early Corticosteroids on 60-day Mortality in Critically Ill Patients with COVID-19: A Multicenter Cohort Study of the OUTCOMEREA Network

PONE-D-21-11495R1

Dear Dr. Dupuis,

We’re pleased to inform you that your manuscript has been judged scientifically suitable for publication and will be formally accepted for publication once it meets all outstanding technical requirements.

Kind regards,

Aleksandar R. Zivkovic

Academic Editor

PLOS ONE

---

## [Editor Report · Acceptance letter]

26 Jul 2021

PONE-D-21-11495R1 

Impact of Early Corticosteroids on 60-day Mortality in Critically Ill Patients with COVID-19: A Multicenter Cohort Study of the OUTCOMEREA Network 

Dear Dr. Dupuis:

I'm pleased to inform you that your manuscript has been deemed suitable for publication in PLOS ONE. Congratulations! Your manuscript is now with our production department. 

Kind regards, 

on behalf of

Dr. Aleksandar R. Zivkovic 

Academic Editor

PLOS ONE